# Dual-Feedback Knowledge Retrieval for Task-Oriented Dialogue Systems

**Tianyuan Shi[1], Liangzhi Li[2*], Zijian Lin[1], Tao Yang[1], Xiaojun Quan[1*], Qifan Wang[3]**

[1]School of Computer Science and Engineering, Sun Yat-sen University, China
[2]Meetyou AI Lab, [3]Meta AI
{shity6, linzj27, yangt225}@mail2.sysu.edu.cn,
quanxj3@mail.sysu.edu.cn, liliangzhi@xiaoyouzi.com,
wqfcr@fb.com

## Abstract

Efficient knowledge retrieval plays a pivotal role in ensuring the success of end-to-end task-oriented dialogue systems by facilitating the selection of relevant information necessary to fulfill user requests. However, current approaches generally integrate knowledge retrieval and response generation, which poses scalability challenges when dealing with extensive knowledge bases. Taking inspiration from open-domain question answering, we propose a retriever-generator architecture that harnesses a retriever to retrieve pertinent knowledge and a generator to generate system responses. Due to the lack of retriever training labels, we propose relying on feedback from the generator as pseudo-labels to train the retriever. To achieve this, we introduce a dual-feedback mechanism that generates both positive and negative feedback based on the output of the generator. Our method demonstrates superior performance in task-oriented dialogue tasks, as evidenced by experimental results on three benchmark datasets. Our code is available at https://github.com/Stycoo/Dual-Feedback-TOD.

## 1 Introduction

Task-oriented dialogue (TOD) systems (Eric et al., 2020) are designed to fulfill specific tasks, such as hotel bookings, through natural language conversations with users. These systems can be integrated into applications such as chatbots and voice assistants, serving various industries like hospitality, e-commerce, and customer service. To generate informative system responses, TOD systems typically rely on an external knowledge base (KB) to retrieve relevant entity information. While large language models (LLMs) like ChatGPT have demonstrated impressive capabilities in understanding multi-turn dialogues and generating fluent responses, there are still cases where they require access to localized KBs to handle specific tasks. Therefore, knowledge

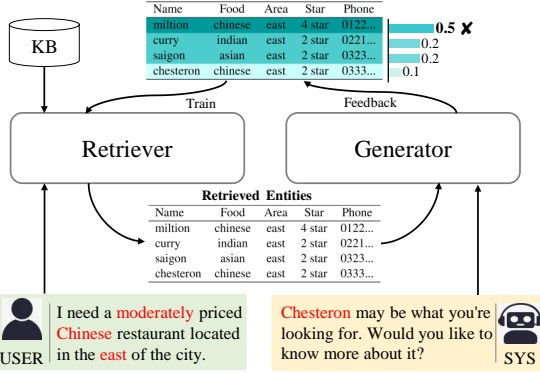

Figure 1: Visualization of erroneous feedback from the generator for retriever training. Entities with darker colors indicate higher relevance scores. The generator mistakenly identifies "Milton" as the most relevant entity, whereas the correct entity should be "Chesterton".

retrieval remains a critical component that necessitates long-term research in dialogue systems.

Traditional pipeline approaches in TOD systems involve multiple modules such as dialogue state tracking and dialogue policy learning, which heavily rely on annotated belief states for knowledge retrieval (Lei et al., 2018; Yang et al., 2021). In contrast, end-to-end task-oriented dialogue (E2E-TOD) systems aim to generate responses in a single step without the need for intermediate retrieval annotations, thereby highlighting the importance of external knowledge retrieval. Existing E2E-TOD systems can be classified into three categories based on their utilization of external knowledge. Firstly, memory networks are employed to store the knowledge, and multi-hop interactions are designed to aggregate relevant information (Madotto et al., 2018; Qin et al., 2020; Raghu et al., 2021). Secondly, pre-trained language models encode linearized KB records, which are then used as input for the response generator (Xie et al., 2022; Wu et al., 2022; Tian et al., 2022). Thirdly, the knowledge base can be embedded into model parameters through data augmentation, enabling implicit

---

*Corresponding authors.

knowledge retrieval (Madotto et al., 2020; Huang et al., 2022). These approaches typically integrate the processes of knowledge retrieval and response generation and train them under the supervision of reference responses. However, this approach suffers from two notable shortcomings. Firstly, the system response often comprises both pure language tokens and KB-related tokens (e.g., hotel names and addresses), making it challenging to train an effective retriever using weak supervision signals from reference responses. Secondly, the efficiency of these systems may decrease as the knowledge base expands in size.

Unlike the aforementioned works, we employ a retriever-generator architecture that explicitly separates the retrieval process from response generation. The retriever is responsible for identifying relevant information from the KB, while the generator utilizes the dialogue context and retrieved information to generate the response. Although this architecture seems straightforward, constructing an effective retriever presents significant challenges, particularly in two key aspects. Firstly, TOD systems inherently possess ground-truth responses to train the generator, but lack annotations for training the retriever. Thus, the retriever can only be trained using weak supervision signals derived from the response generator. Secondly, within a specific domain, different entities often exhibit structural and content similarities, making it challenging for the generator to learn which entities are truly relevant. Consequently, the weak supervision signals from the generator may not be reliable. Figure 1 provides a visual illustration of these challenges.

To tackle these challenges, we propose a dual-feedback mechanism for the retriever, consisting of positive feedback and negative feedback. Positive feedback is constructed based on the conditional generation probabilities of responses corresponding to different retrieved entities. Intuitively, if the relevance between an entity and the response is higher, the conditional generation probability of the response corresponding to this entity will also be higher. Utilizing positive feedback allows us to train the retriever based on the knowledge learned by the generator from reference responses. In order to prevent the retriever from being misled by inaccurate information, we contend that calibration is necessary. Calibration involves the initial identification and resolution of errors, which we accomplish by sampling negative samples derived from the generator's hypothesis responses. Then, we construct negative feedback based on these negative samples to calibrate the positive feedback.

We conducted evaluations of our system on three well-established TOD datasets: Multi-WOZ 2.1 (MWOZ) (Eric et al., 2020), Stanford Multi-Domain dataset (SMD) (Eric et al., 2017), and CamRest (Wen et al., 2017). The experimental results demonstrate that our model outperforms the baseline methods, particularly in scenarios involving large-scale KB. Additionally, through extensive analysis, we have made several notable findings. Firstly, our retriever exhibits clear advantages over the baseline methods as the size of KB increases. Secondly, our dual-feedback mechanism effectively mitigates the problem of incorrect knowledge learned solely with positive feedback from the generator. Lastly, our method exhibits good compatibility with LLMs like ChatGPT.

## 2 Related Work

### 2.1 End-to-End Task-Oriented Dialogue

E2E-TOD systems employ different strategies to incorporate KB information for response generation. Firstly, memory networks are utilized to store knowledge, using multi-hop interactions to aggregate relevant information. For instance, Mem2seq (Madotto et al., 2018) employs multi-hop attention over memory cells to select KB-related tokens during response generation. GLMP (Wu et al., 2019) introduces a global-to-local memory pointer network to retrieve relevant triplets and complete the response template. CD-NET (Raghu et al., 2021) retrieves relevant KB records by computing a distillation distribution based on dialogue context.

Secondly, the entire linearized KB is encoded by pre-trained language models and taken as input to a generator to generate the final system response. For instance, UnifiedSKG (Xie et al., 2022) uses a unified text-to-text framework, while Q-TOD (Tian et al., 2022) rewrites the dialogue context into a natural language query for knowledge retrieval. MAKER (Wan et al., 2023) introduces a multi-grained retrival with both entity and attribute selection.

Thirdly, knowledge bases are stored in model parameters for implicit retrieval during response generation. GPT-KE (Madotto et al., 2020) embeds the KB into pre-trained model parameters through data augmentation. Following GPT-KE, ECO (Huang et al., 2022) first generates the most relevant entity

with trie constraint before response generation to ensure entity consistency in the response.

## 2.2 Knowledge Retrieval

Previous research has extensively explored methods for knowledge retrieval across various tasks, including question answering (Chen et al., 2017; Kwiatkowski et al., 2019), fact checking (Thorne et al., 2018), and dialogue systems (Dinan et al., 2019). Recently, neural network-based dense retrievers have become popular. These retrievers commonly use a dual-encoder architecture (Yih et al., 2011), where queries and passages are encoded as separate vectors, and relevance is determined through inner product or Euclidean distance.

Supervised retrievers, such as DPR (Karpukhin et al., 2020), have been developed for open-domain question answering. These retrievers are trained using labeled question-document pairs. To overcome the need for costly query-document annotations, researchers have explored alternative approaches that leverage signals from the answer generator. REALM (Guu et al., 2020) and RAG (Lewis et al., 2020) propose joint training of the retriever and the generator by treating documents as latent variables. FiD-KD (Izacard and Grave, 2021a) employs cross-attention scores as supervision through knowledge distillation. EMDR$^2$ (Sachan et al., 2021) models retrieval decisions as latent variables and employs an expectation-maximization algorithm to approximate the computation. Unsupervised approaches have also been explored. ICT (Lee et al., 2019) introduces the inverse cloze task for unsupervised pre-training of dense retrievers. Izacard et al. (2022) investigate contrastive learning methods for training retrievers, while Spider (Ram et al., 2022) utilizes recurring spans within a document to create pseudo-positive query-document pairs.

## 3 Methods

As illustrated in Figure 2, our system comprises a knowledge retriever and a response generator. The retriever first retrieves top-$K$ relevant entities from a knowledge base. These retrieved entities, along with the dialogue context, are then fed into the generator model to generate the response.

### 3.1 Notations

Given a dialogue $\mathcal{D} = \{u_1, y_1, ..., u_T, y_T\}$ consisting of $T$ turns, where $u_t$ and $y_t$ represent the user utterance and system response at the $t$-th turn,

respectively, we denote the dialogue context of the $t$-th turn as $c_t = \{u_1, y_1, ..., u_{t-1}, y_{t-1}, u_t\}$. An external KB is provided, represented as a set of entities, i.e., $\mathcal{E} = \{e_1, e_2, ..., e_B\}$, where each entity $e_i$ consists of $N$ attribute-value pairs, i.e., $e_i = \{a^1, v_i^1, ..., a^N, v_i^N\}$. E2E-TOD takes the dialogue context $c_t$ and the KB as input and directly generates a natural language response $y_t$.

## 3.2 Knowledge Retriever

Our knowledge retriever comprises an encoder $\text{Enc}_r$ that maps any input to a $d$-dimensional vector. The user utterances and system responses in the dialogue context $c_t$ are first concatenated and encoded by $\text{Enc}_r$ as the query. To encode an entity, we concatenate the attribute-value pairs of the entity into a sequence and pass it to $\text{Enc}_r$. The similarity score between $c_t$ and $e_i$ is computed by taking the dot product of their respective vectors:

$$s_{t,i} = \text{Enc}_r(c_t)^T \text{Enc}_r(e_i). \quad (1)$$

Based on these similarity scores, we identify the top-$K$ entities from the entity set $\mathcal{E}$ as the candidate entities for response generation. This set of candidates is represented as $\hat{\mathcal{E}} = \{e_1, e_2, ..., e_K\}$.

We employ BERT (Devlin et al., 2019) as the encoder and extract the representations of the [CLS] token to represent $c_t$ and $e_i$. Prior research has highlighted that initializing the encoder directly with BERT weights can lead to collapsed representations and impact retrieval performance. Consequently, we initialize the weights via pre-training with distant supervision (Qin et al., 2019).[1]

## 3.3 Response Generator

Our generator is built upon the Fusion-in-Decoder (FiD) (Izacard and Grave, 2021b) model, which is based on the pre-trained T5 (Raffel et al., 2020). The FiD model consists of an encoder $\text{Enc}_g$ and a decoder $\text{Dec}_g$. The encoder is responsible for processing $K$ different text inputs independently, where each input $x_{t,i}$ is formed by concatenating the dialogue context $c_t$ and a candidate entity $e_i$. The output representations of the encoder are then concatenated to create a global representation $X_t = [\text{Enc}_g(x_{t,1}), ..., \text{Enc}_g(x_{t,K})]$ for the current turn.

The decoder takes $X_t$ as input and generates the response autoregressively. During this process, the decoder utilizes causal attention to incorporate information from previously generated tokens, as

---

[1] Details of the pre-training are available in Appendix D.

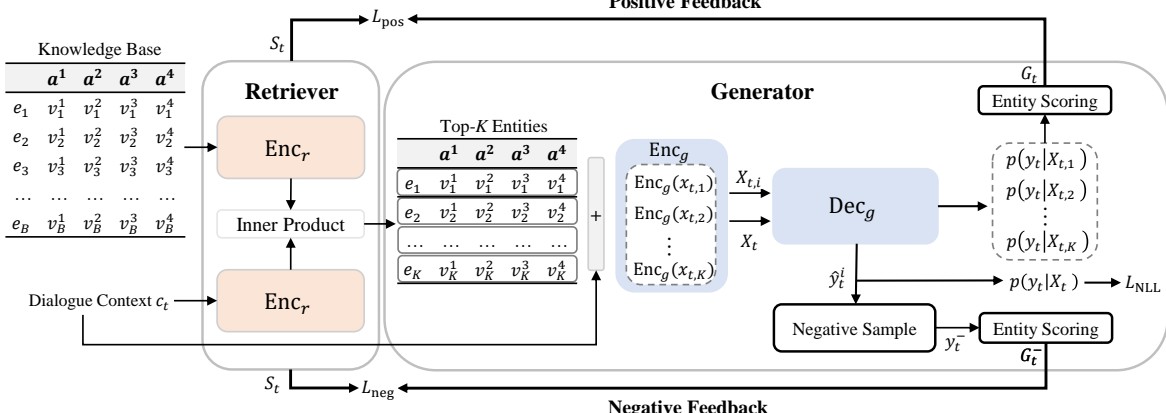

Figure 2: The overall architecture of our end-to-end task-oriented dialogue system, comprising a knowledge retriever and a response generator. The retrieval component is trained using dual feedback from the response generator.

well as cross-attention to the input tokens represented by $X_t$. This enables the decoder to consider information from both the generated tokens and the tokens of retrieved entities. The probability of the response is defined as follows:

$$p(y_t|X_t) = \prod_{i=1}^{|y_t|} p(y_{t,i}|y_{t,<i}, X_t), \qquad (2)$$

where $|y_t|$ represents the length of the response $y_t$.

### 3.4 End-to-End Training

**Entity scoring** Given the retrieved entities $\hat{\mathcal{E}} = \{e_1, e_2, ..., e_K\}$, we employ the aforementioned response generator to assign a score to each entity. Firstly, the dialogue context $c_t$ and each entity $e_i$ are concatenated to form $x_{t,i}$, which is then fed into the generator. The conditional log probability (length-normalized) of the response $y_t$ is utilized to score entity $e_i$ as follows:

$$\begin{aligned} g_{t,i} &= p(y_t|X_{t,i}) \\ &= \frac{\sum_{i=1}^{|y_t|} \log p(y_{t,i}|y_{t,<i}, X_{t,i})}{|y_t|}, \end{aligned} \qquad (3)$$

where $X_{t,i} = \text{Enc}_g(x_{t,i})$.

The rationale behind the scoring is straightforward: if an entity is pertinent to the response, the probability of generating this response given the dialogue context and the entity should also be high.
**Positive feedback** We utilize the entity scores $G_t = \{g_{t,i}\}_{1 \le i \le K}$ obtained from the generator as pseudo-labels to train the retriever, with the objective of transferring the knowledge acquired by the generator from reference responses to the retriever. To achieve this, we enforce consistency

between the retrieval scores $S_t = \{s_{t,i}\}_{1 \le i \le K}$ obtained from Eq. (1) and the pseudo-labels $G_t$ using KL-divergence as follows:

$$\begin{aligned} L_{\text{pos}} &= D_{KL}(G_t, S_t) \\ &= \sum_{i=1}^{K} \tilde{g}_{t,i}(\log \tilde{g}_{t,i} - \log \tilde{s}_{t,i}), \end{aligned} \qquad (4)$$

where $\tilde{g}_{t,i} = \text{softmax}(G_t)_i$, $\tilde{s}_{t,i} = \text{softmax}(S_t)_i$.

The use of the generator's supervision to train the retriever is referred to as *positive feedback*. However, it is important to note that the generator may assimilate incorrect knowledge from the entities, leading to inaccurate entity relevancy scores.

**Negative feedback** To address the issue of positive feedback when training the retriever, we propose incorporating negative feedback from the generator to calibrate the pseudo-labels. To achieve this, we select a negative sample for the global representation $X_t$ during the generation process. A negative sample refers to a response that exhibits a high generation probability but is of low quality. Specifically, we generate a set $\{\hat{y}_t^i\}_{1 \le i \le M}$ of responses using beam search, where $M$ is the beam search size, and rank them based on their generation probabilities. We use $R_t^g(\hat{y}_t^i)$ to represent the rank of candidate response $\hat{y}_t^i$. Note that a high generation probability does not guarantee a high-quality response. To evaluate the quality, we define a function $o(\hat{y}_t^i, y_t)$ that measures the overlap between a response and the reference response. We implement $o(\hat{y}_t^i, y_t)$ using the BLEU metric. As a result, we obtain a new sorted list of responses based on their true quality, where the rank of candidate response $\hat{y}_t^i$ is $R_t^o(\hat{y}_t^i)$. Finally, we identify

the negative sample as follows:

$$y_t^- = \mathrm{argmin}_{\hat{y}_t^i}(R_t^g(\hat{y}_t^i) - R_t^o(\hat{y}_t^i)). \quad (5)$$

Intuitively, it is desirable for the retriever to avoid incorporating the signal from the negative sample when updating its parameters. Therefore, we employ Eq. (3) to calculate the score of each entity in generating the negative sample, and utilize the scores $G_t^- = \{g_{t,i}^-\}_{1 \le i \le K}$ for all retrieved entities to calibrate the positive feedback by introducing a margin loss:

$$L_{\mathrm{neg}} = \max\left(0, -(D_{\mathrm{KL}}(G_t^-, S_t) - D_{\mathrm{KL}}(G_t, S_t)) + \eta\right). \quad (6)$$

The training objective of the retriever comprises $L_{\mathrm{pos}}$ and $L_{\mathrm{neg}}$. Additionally, we train the response generator by minimizing the negative log-likelihood loss:

$$L_{\mathrm{NLL}} = -\sum_{i=1}^{|y_t|} \log p(y_{t,i}|y_{t,\le i}, X_t). \quad (7)$$

The final training objective is a combination of the objectives of the retriever and the generator:

$$L = L_{\mathrm{NLL}} + L_{\mathrm{pos}} + L_{\mathrm{neg}}. \quad (8)$$

## 4 Experiments

In this section, we provide a detailed description of the experiment and present the main results.

### 4.1 Datasets

We conduct our evaluation using three publicly available TOD datasets: Multi-WOZ 2.1 (MWOZ) (Eric et al., 2020), Stanford Multi-Domain dataset (SMD) (Eric et al., 2017), and CamRest (Wen et al., 2017). These datasets contain dialogues that are associated with relevant KBs. This is referred to as the session-level KB. To construct a comprehensive and extensive knowledge base, we merge the session-level KBs corresponding to each dialog turn, resulting in a dataset-level KB. The provided train, validation, and test splits for each dataset are utilized in our evaluation. The statistics of the three datasets are summarized in Appendix A.

### 4.2 Evaluation Metrics

We evaluate the performance of our model using two metrics. Firstly, we calculate the top-$K$ retrieval recall (Re@$K$), which is inspired by the widely used top-$K$ retrieval accuracy in question

answering (Karpukhin et al., 2020). Re@$K$ measures the effectiveness of the retriever by determining the percentage of gold attribute values covered by the entities retrieved in the top-$K$ list. Secondly, we assess the overall performance of our TOD system from two aspects. To evaluate the system's capability to generate relevant entities, we employ the micro Entity-F1 metric (Eric et al., 2017). Additionally, we utilize the BLEU metric to measure the N-gram overlap between the generated response and the reference response. With these metrics, we can comprehensively evaluate the performance of our model in both retrieval and TOD systems.

### 4.3 Experimental Settings

We instantiate the knowledge retriever using the BERT-base model. As for the generator, we instantiate it with T5 of varying model sizes: T5-base and T5-large. Both the retriever and generator models are fine-tuned using the ADAM algorithm (Kingma and Ba, 2015) with different learning rate schedulers. The retriever model employs a fixed learning rate scheduler, while the generator model uses a linear learning rate scheduler. The experiments are conducted on a single 24G NVIDIA RTX 3090 GPU. We select the checkpoint that yields the best results on the validation set. For more detailed information regarding our experimental setup, please refer to Appendix B.

### 4.4 Baselines

In our comparison, we classify the existing approaches for E2E-TOD systems into three categories based on their utilization of KB.

**Memory networks:** These methods store external knowledge in memory cells in the form of triplets and utilize multi-hop attention to retrieve relevant information for generating responses. Examples include DSR (Wen et al., 2017), KB-Retriever (Qin et al., 2019), GLMP (Wu et al., 2019), DF-Net (Qin et al., 2020), FG2Seq (He et al., 2020), and CDNET (Raghu et al., 2021).

**Linearized KB:** These approaches leverage pre-trained language models to encode the entire linearized KB, along with the dialogue context, as input for assisting response generation. Examples include DialoKG (Rony et al., 2022), UnifiedSKG (Xie et al., 2022), and Q-TOD (Tian et al., 2022).

**Model parameters:** These approaches encode the KB into model parameters through data augmentation of the dialogues with KB records, enabling implicit retrieval during response genera-

| Model | MWOZ | | SMD | | CamRest | |
|---|---|---|---|---|---|---|
| | **BLEU** | **Entity-F1** | **BLEU** | **Entity-F1** | **BLEU** | **Entity-F1** |
| GLMP | 6.90† | 32.40† | 13.90† | 60.70† | 15.10 | 58.90 |
| DF-Net | 9.40 | 35.10 | 14.40 | 62.70 | - | - |
| GPT2-KE | 9.40 | 35.10 | 14.40 | 62.70 | - | - |
| FG2Seq | 14.60* | 36.50* | 16.80* | 61.10* | 20.20* | 66.40* |
| CDNET | 11.90 | 38.70 | 17.80 | 62.90 | 21.80 | 68.60 |
| DialogKG | 12.60 | 43.50 | 20.00 | 65.90 | 23.40 | **75.60** |
| UnifiedSKG (T5-large) | 13.69◇ | 46.04◇ | 17.27◇ | 65.85◇ | 20.31◇ | 71.03◇ |
| Q-TOD (T5-base) | - | - | 20.14 | 68.22 | - | - |
| Q-TOD (T5-large) | 17.62 | 50.61 | 21.33 | 71.11 | 23.75 | 74.22 |
| Ours (T5-base) | 18.26 | 52.52 | 24.12 | 69.36 | 25.85 | 72.83 |
| Ours (T5-large) | **18.48** | **53.17** | **25.10** | **71.58** | **26.00** | 74.04 |

Table 1: Main results of E2E-TOD systems with session-level KBs on MWOZ, SMD, and CamRest, with the best scores highlighted in bold. †, ∗, and ◇ indicate that the results are sourced from (Qin et al., 2020), (Raghu et al., 2021), and (Tian et al., 2022), respectively.

tion. Examples include GPT-2+KE (Madotto et al., 2020) and ECO (Huang et al., 2022).

| Model | MWOZ | | CamRest | |
|---|---|---|---|---|
| | **BLEU** | **Entity-F1** | **BLEU** | **Entity-F1** |
| DF-Net | 6.45 | 27.31 | - | - |
| FG2Seq | 10.74 | 33.68 | 19.20 | 59.35 |
| CDNET | 10.90 | 31.40 | 16.50 | 63.60 |
| Q-TOD (T5-large) | 16.67 | 47.13 | 21.44 | 63.88 |
| Ours (T5-base) | 17.61 | 51.61 | **27.39** | 70.74 |
| Ours (T5-large) | **18.36** | **52.96** | 26.61 | **73.58** |

Table 2: Main results with dataset-level KBs on MWOZ and CamRest. Best scores are highlighted in bold.

## 4.5 Main Results

We conducted experiments in both session-level and dataset-level KB scenarios. In the following paragraphs, we discuss the detailed results.

**Session-level KB** The results for the session-level KB setting are summarized in Table 1. Our system, instantiated with T5-large, achieves state-of-the-art performance on the MWOZ and SMD datasets. Specifically, our method demonstrates an improvement of 2.56 in the Entity-F1 metric for MWOZ and 0.47 for SMD over Q-TOD. Moreover, our system achieves the highest BLEU scores, with an increase of 0.86 on MWOZ, 3.77 on SMD, and 2.25 on CamRest compared to Q-TOD. However, our system does not achieve the best performance in terms of Entity-F1 on the CamRest dataset. This can be attributed to the presence of session-level KBs containing only 1-2 entities, which poses a challenge for the retriever to perform optimally.

Note that Q-TOD also employs a Transformer-based response generator similar to ours and utilizes manually annotated queries. However, our

method outperforms Q-TOD on all three datasets using T5-large. We believe this is because Q-TOD trains the retriever independently, whereas our proposed dual-feedback method allows for joint training of the retriever and generator. This facilitates better alignment between the retrieved entities and their relevance to the current dialogue context.

**Dataset-level KB** Table 2 presents the results of our system on the dataset-level KB and compares it with the reimplementation of several well-known E2E-TOD systems. Our system demonstrates significant advantages, particularly in the Entity-F1 metric. It achieves an improvement of 5.83 on the MWOZ dataset and 9.7 on the CamRest dataset over the strong Q-TOD baseline.

Furthermore, when comparing the experimental results in the two KB settings, we observe that our model exhibits more stable performance. As the KB size increases, our model experiences only a minor decrease in performance, while other baseline models show a noticeable decline. For instance, DF-Net exhibits a decrease of 7.79 in terms of Entity-F1 on the MWOZ dataset, indicating that the method struggles to adapt to large-scale KBs. Similarly, Q-TOD experiences a reduction of 3.48 in Entity-F1 on MWOZ, highlighting the less stable performance of its independently trained retriever.

## 5 Analysis and Discussion

To further investigate our method, we conduct a comprehensive analysis that includes an ablation study, an exploration of different retriever training methods, an examination of different negative sample selecting methods, and an assessment of

| Method | Dataset-level KB | | | Session-level KB | | |
|---|---|---|---|---|---|---|
| | BLEU | Entity-F1 | Re@7 | BLEU | Entity-F1 | Re@3 |
| *Ablation* | | | | | | |
| Ours | **17.61** | **51.61** | **90.98** | **18.26** | **52.52** | **79.26** |
| *w/o* negative feedback | 17.54 | 50.32 | 87.97 | 17.13 | 51.49 | 72.93 |
| *w/o* positive feedback | 16.05 | 48.07 | 83.46 | 17.64 | 50.48 | 76.69 |
| *Retriever training methods* | | | | | | |
| Attention-based | 16.42 | 49.44 | 85.71 | 16.61 | 51.16 | 69.17 |
| *w/* negative feedbcak | 16.83 | 50.23 | 89.47 | 16.73 | 52.08 | 77.78 |
| MML-based | 17.24 | 49.26 | 84.15 | 17.47 | 51.07 | 70.05 |

Table 3: Results of ablation study and different retriever training methods, where "*w/*" means "with" and "*w/o*" means "without". "Attention-based" and "MML-based" indicate the attention-based (Izacard and Grave, 2021a) and the maximum marginal likelihood (MML)-based (Sachan et al., 2021) retriever training methods, respectively.

compatibility with LLMs. T5-base is used as the base model for the first three analyses.

## 5.1 Ablation Study

We conduct an ablation study on the MWOZ dataset to evaluate our method under both session-level KB and dataset-level KB settings. In the session-level KB setting, we adjust the retriever evaluation metric from Re@7 to Re@3 due to the smaller size of the KB. The results of this study are presented in the upper section of Table 3.

By removing negative feedback, we observe a decrease in retrieval metrics (Re@7/Re@3) and TOD metrics (BLEU/Entity-F1). Notably, the retrieval metrics exhibit a more pronounced decline. This suggests that the incorporation of negative feedback allows the retriever to more effectively learn from the generator, resulting in more relevant entities for response generation.

Furthermore, the further removal of positive feedback reduces the model to a learnable generator with a fixed retriever. In the dataset-level KB setting, all evaluated metrics exhibit a further decline. This indicates that our positive feedback successfully facilitates the transfer of knowledge learned by the generator to the retriever.

However, in the session-level KB setting, while the Entity-F1 metric decreases, the Re@3 metric increases. We attribute this to the small size of the KB, causing significant fluctuations in the top-3 entities as the retriever updates. Consequently, the generator tends to prioritize the entities that occur most frequently among the retrieved results. While this improves the model's performance in dialogue scenarios that only require a single entity, which constitutes the majority, it hampers performance in scenarios that require multiple entities.

## 5.2 Methods for Retriever Training

We conduct a comparative experiment with other retriever training methods commonly used in question answering (QA), including attention-based (Izacard and Grave, 2021a) and maximum marginal likelihood (MML)-based (Sachan et al., 2021) methods. To further demonstrate the necessity of negative feedback, we perform additional experiments by incorporating negative feedback into the attention-based method. Note that the underlying model architecture remains the same across all methods, with the difference lying solely in the employed training methods. The results are presented in the lower section of Table 3. Due to the similar performance trends observed for both dataset-level and session-level KB settings, the following analysis will focus solely on the dataset-level KB setting.

We observe that our proposed method outperforms the attention-based method by 2.26 and the MML method by 3.82 in terms of the Re@7 metric. This indicates that in the TOD task where KB-related tokens are sparse in the response, our utilization of conditional probability of response enables better correlations between the retrieved entities and the response. Furthermore, when comparing the performance of the attention-based method before and after incorporating negative feedback, we consistently observe improvements across all metrics. This suggests that even in existing methods, negative feedback can effectively enhance the performance of the retriever and the overall model.

## 5.3 Methods for Selecting Negative Sample

To understand how different negative sample selecting methods affect the results, we conduct additional experiments. As depicted in Figure 3, these methods can be categorized into two types: $argmin_{***}$ and $rank_{***}$. The $argmin_{***}$ method re-

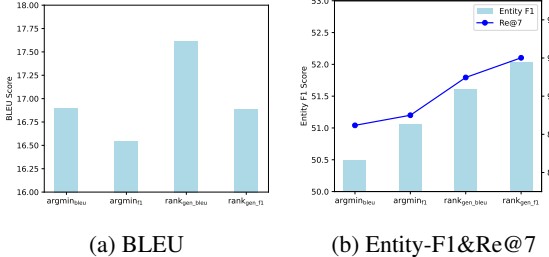

(a) BLEU        (b) Entity-F1&Re@7

Figure 3: Results of different negative sample selecting methods on MWOZ, where argmin$_{***}$ means selecting the response with the lowest score for negative feedback, and rank$_{***}$ combines the scoring of the oracle function with the generation probabilities to select the response.

lies solely on the scoring of the oracle function to select the response with the lowest score to construct negative feedback. On the other hand, the rank$_{***}$ method combines the scoring of the oracle function with the generation probabilities to select responses with high probabilities but low factual quality to construct negative feedback. Additionally, we also experiment with using BLEU and Entity-F1 as the oracle functions to assess their influence on the experiments. From Figure 3, we have made three notable observations.

Firstly, the results in Entity-F1 and Re@7 demonstrate consistent trends, indicating that improvements in retrieval performance lead to enhanced performance in TOD. This highlights the significance of enhancing the retrieval performance. Secondly, the rank$_{***}$ methods outperform the argmin$_{***}$ methods. Therefore, we conclude that the rank$_{***}$ methods are more accurate in selecting appropriate negative samples that could reflect the generator's mistakes compared to methods that solely rely on the oracle function.

Lastly, it is observed that using Entity-F1 as the oracle function often results in higher Entity-F1 and Re@7 scores, but slightly lower BLEU scores. This suggests that Entity-F1 can provide more precise feedback regarding entity selection compared to BLEU. However, in practical scenarios, obtaining gold entity annotations from responses is challenging, despite their availability in the MWOZ dataset. This is why we employ BLEU as the oracle function in our experiments.

### 5.4 Compatibility with LLMs

To demonstrate the compatibility of our proposed dual-feedback mechanism with LLMs, we conduct zero-shot and few-shot experiments using ChatGPT

as the generator. We train the retriever using positive and negative feedback provided by ChatGPT. The experimental results are presented in Table 4. Notably, since the ChatGPT API [2] does not directly provide the generation probabilities of responses, we make slight adaptations. The details of this experiment can be found in Appendix F.

From Table 4, we observe that the best performance is achieved when incorporating negative feedback in both the zero-shot and few-shot scenarios. This highlights the effectiveness of the dual-feedback mechanism in training a superior retriever, even when utilizing LLMs. Moreover, the performance in the few-shot scenario surpasses that in the zero-shot scenario, emphasizing the importance of having a few demonstrating examples. These examples provide valuable guidance and enable the model to better understand the task.

| Task | BLEU | Entity-F1 | Re@7 |
|---|---|---|---|
| Zero-shot | 5.95 | 27.44 | 83.46 |
| *w/* positive feedback | 6.21 | 28.97 | 85.76 |
| *w/* negative feedback | **6.98** | **30.48** | **87.97** |
| Few-shot | 6.78 | 28.73 | 83.46 |
| *w/* positive feedback | 6.43 | 29.07 | 84.49 |
| *w/* negative feedback | **7.14** | **31.46** | **88.03** |

Table 4: Results of employing ChatGPT as the generator on the MWOZ dataset.

### 6 Conclusion

In this paper, we propose a novel dual-feedback knowledge retriever for E2E-TOD systems. Our approach separates the knowledge retrieval process from response generation, and leverages the knowledge learned by the generator to create synthetic positive and negative feedback for retriever training, eliminating the need for retrieval annotations. Through empirical evaluations, we demonstrate that our system achieves state-of-the-art performance, regardless of whether a small or large-scale KB is used in each dialogue. Furthermore, ablation studies indicate that our dual-feedback mechanism effectively mitigates the problem of incorrect knowledge learned solely from positive feedback generated by the generator. Consequently, this improvement in retrieval performance directly translates to enhanced performance in E2E-TOD systems. Lastly, our method exhibits good compatibility with LLMs like ChatGPT.

---

[2]https://openai.com/api/

## Limitations

There are three potential limitations to consider in our work. Firstly, the process of obtaining negative feedback through response sampling can result in decreased training efficiency. Secondly, training the retriever using feedback from ChatGPT can be costly. In each epoch, it is necessary to re-predict for every sample, and finding ways to reuse intermediate predictions is an area to explore. Thirdly, there still exists a noticeable gap between fine-tuning the generator and achieving few-shot learning with ChatGPT. Future research is needed to investigate methods for narrowing this gap.

## Ethics Statement

All experiments in this study were conducted using publicly available datasets that do not contain any private information. Our work does not involve the analysis or utilization of identity characteristics, and we do not engage in any form of gender or racial discrimination.

## Acknowledgements

This work was supported by the National Natural Science Foundation of China (No. 62176270), the Guangdong Basic and Applied Basic Research Foundation (No. 2023A1515012832).

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

## A Dataset Statistics

In the case of MWOZ, each dialogue session includes 7 candidate entities, while for SMD and CamRest, the sizes of the candidate KBs vary, ranging from 0 to 8 and 0 to 57, respectively. Table 5 shows the statistics of the three datasets: Multi-WOZ 2.1 (MWOZ) (Eric et al., 2020), Stanford Multi-Domain dataset (SMD) (Eric et al., 2017), and CamRest (Wen et al., 2017). When we include the entire knowledge base (KB) as input, the length of the input text becomes significantly long, posing challenges for existing end-to-end task-oriented dialogue systems to handle.

| Dataset | Statistics | | | Sequence Length | |
|---|---|---|---|---|---|
| | #Dial | #Utt | Dial | *w/* SessionKB | *w/* DatasetKB |
| MWOZ | 2877 | 19870 | 730 | 996 | 23730 |
| SMD | 3031 | 15928 | 109 | 435 | - |
| CamRest | 676 | 2744 | 156 | 393 | 1356 |

Table 5: Dataset statistics. We count the maximum input lengths for different cases: dialogue only (Dial), dialogue with session-level KB (*w/* SessionKB), and dialogue with dataset-level KB (*w/* DatasetKB).

## B Hyperparameters

The hyperparameters of our system with session-level and dataset-level KB are shown in Table 6 and Table 7, respectively.

## C Traing Process

We provide a more comprehensive description of the training process below. During the initial training phases, we incorporate a warm-up period for the generator, facilitating dependable feedback for the retriever. Subsequent to this phase, both the retriever and generator are subjected to joint training until reaching convergence. It's important to

| Parameters | MWOZ | | SMD | | CamRest | |
|---|---|---|---|---|---|---|
| | T5-base | T5-large | T5-base | T5-large | T5-base | T5-large |
| Retriever LR | 2e-5 | 2e-5 | 2e-5 | 2e-5 | 2e-5 | 2e-5 |
| Retriever LR schedule | Fixed | Fixed | Fixed | Fixed | Fixed | Fixed |
| Retriever input max length | 128 | 128 | 128 | 128 | 128 | 128 |
| Top-K retrieved entities | 6 | 6 | 6 | 6 | 5 | 4 |
| Retriever training start step | 625 | 625 | 750 | 750 | 750 | 750 |
| Generator LR | 1e-4 | 1e-4 | 1e-4 | 1e-4 | 1e-4 | 1e-4 |
| Generator LR schedule | Linear | Linear | Linear | Linear | Linear | Linear |
| Generator input context max length | 200 | 200 | 200 | 200 | 200 | 200 |
| Generator input KB max length | 100 | 100 | 200 | 200 | 200 | 200 |
| Response max length | 64 | 64 | 128 | 128 | 64 | 64 |
| Beam search size | 5 | 5 | 5 | 5 | 5 | 5 |
| Oracle function | BLEU | BLEU | BLEU | BLEU | BLEU | BLEU |
| Batch size | 2 | 1 | 2 | 2 | 2 | 1 |
| Training steps | 1500 | 1500 | 1500 | 1500 | 1000 | 1500 |
| Grad accumulation steps | 32 | 64 | 32 | 32 | 32 | 32 |

Table 6: Hyperparameter settings of our system when session-level KBs are used on MWOZ, SMD and CamRest.

| Parameters | MWOZ | | CamRest | |
|---|---|---|---|---|
| | T5-base | T5-large | T5-base | T5-large |
| Retriever LR | 2e-5 | 2e-5 | 2e-5 | 2e-5 |
| Retriever LR schedule | Fixed | Fixed | Fixed | Fixed |
| Retriever input max length | 128 | 128 | 128 | 128 |
| Top-K retrieved entities | 10 | 10 | 10 | 7 |
| Generator LR | 1e-4 | 1e-4 | 1e-4 | 1e-4 |
| Generator LR schedule | Linear | Linear | Linear | Linear |
| Generator input context max length | 200 | 200 | 200 | 200 |
| Generator input KB max length | 100 | 100 | 200 | 200 |
| Response max length | 64 | 64 | 64 | 64 |
| Beam search size | 5 | 5 | 5 | 5 |
| Retriever training start step | 750 | 750 | 750 | 750 |
| Oracle function | BLEU | BLEU | BLEU | BLEU |
| Batch size | 2 | 1 | 2 | 1 |
| Training steps | 1500 | 1500 | 1000 | 1500 |
| Grad accumulation steps | 32 | 64 | 32 | 32 |

Table 7: Hyperparameter settings of our system when the dataset-level KBs are used on MWOZ and CamRest.

emphasize that we carry out negative sampling at each training step, ensuring consistent information updates from the generator to the retriever.

While the inclusion of sampling operations during training does indeed extend the overall training duration, it is important to note that these sampling operations are exclusively carried out within the training phase and, therefore, do not impact the inference time.

## D   Retriever Pre-training

Given a dialogue context and the system response, we utilize the entity with the highest frequency of its attribute values in the dialogue context and system response as the label. To optimize this process, we employ supervised contrastive learning (Gao et al., 2021). Specifically, the positive example for a dialogue context is the corresponding

| Parameters | MWOZ | CamRest |
|---|---|---|
| Batch size | 128 | 108 |
| Epoch | 10 | 15 |
| LR schedule | Linear | Linear |
| LR | 5e-5 | 5e-5 |
| Max input length | 128 | 128 |
| Pooling type | CLS | CLS |
| Weight decay | 0.01 | 0.01 |

Table 8: Hyperparameter setting for pre-training our retriever on the dataset-level KBs of MWOZ and CamRest datasets, respectively.

labeled entity, while the negative examples are the labeled entities from other examples in the same mini-batch. We utilize the InfoNCE loss as the training objective, aiming to bring the sentence representations of positive samples closer together and push away those of negative samples. This pre-training procedure is performed on the MWOZ and CamRest datasets. Since the knowledge base in the SMD dataset is specific to each dialogue and lacks a global knowledge base, we do not conduct pre-training on the SMD dataset. The hyperparameters for the pre-training are detailed in Table 8.

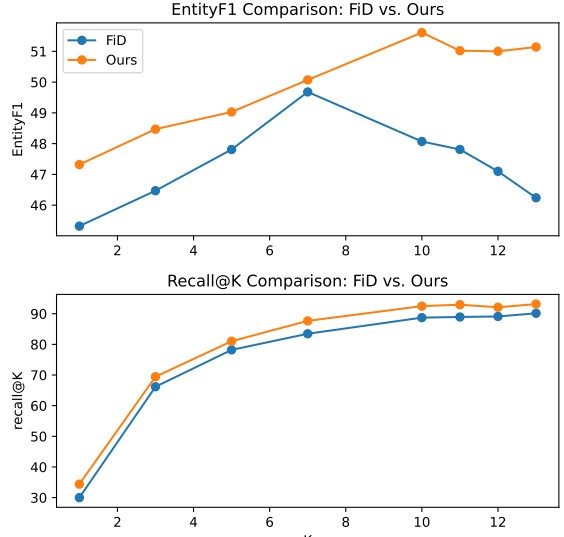

Figure 4: Performance on MWOZ as the number of retrieved entities changes.

## E    The Number of Retrieved Entities

In our experiments, the retriever retrieves the top-$K$ relevant entities from the knowledge base for response generation. We conduct an analysis to assess the performance of our system and FiD as we varied the number of retrieved documents $K$, as shown in Figure 4. Notably, we observe that FiD achieves the best performance when $K = 7$, while our model exhibits optimal performance when $K = 10$. This suggests that a small number of $K$ might be inadequate to cover necessary knowledge entities for response generation. However, as $K$ further increases, FiD's performance is significantly affected, leading to a noticeable decline in performance. In contrast, our system demonstrates only a slight decrease in performance. This finding suggests that a large value of $K$ would inevitably introduce more noisy entities and increase the diffi-

culty of knowledge utilization in response generation. Nevertheless, our model, empowered by the dual-feedback mechanism, still enables effective retriever training even in such circumstances.

## F    Train the Retriever Using ChatGPT

Regarding the zero-shot and few-shot experiments, our retriever underwent training on the complete MultiWOZ dataset, with a total of 2877 dialogues. Furthermore, in the few-shot context, we create prompts to simulate three main knowledge retrieval scenarios, aiming to enhance the model's comprehension of dialogue tasks. These scenarios involve knowledge base (KB) containing entities that align with user intent, KB containing similar entities for recommendations, and KB lacking similar entities and leading to query failures. Furthermore, we have also taken into account the potential inaccuracies in self-reported ChatGPT scores, which could negatively impact the feedback. Our experimental results indicate that the proposed composite scoring strategy, combining self-reported scores and BLEU metrics, effectively alleviates this issue.

### F.1    Hyperparameters

The hyperparameters for our retriever when using ChatGPT as the generator are presented in Table 9.

| Parameters | MWOZ |
|---|---|
| Batch size | 8 |
| Epoch | 5 |
| LR | 5e-5 |
| LR schedule | Fixed |
| Max input length | 128 |
| Pooling type | CLS |
| Weight decay | 0.01 |
| Max output tokens | 150 |
| Response Temperature | 1.0 |
| Rank Temperature | 0.1 |
| Oracle function | BLEU |
| Beam size | 5 |

Table 9: Hyperparameter setting for retriever training when using ChatGPT as the generator, as well as the configuration of ChatGPT, on the dataset-level knowledge base of MWOZ.

### F.2    Accuracy of ChatGPT Scores

As indicated in the table below, relying solely on self-reported (sr) scores for constructing feedback leads to performance degradation compared to the approach proposed in the paper. This, to some extent, validates the presence of inaccuracies in

self-reported scores. However, quantifying the precision is challenging due to the absence of ground truth labels for ChatGPT's outputs. Furthermore, our approach combines self-reported scores with BLEU-based ranking for selecting negative samples, thereby mitigating the unfavorable effects that may arise from these inaccuracies.

| Scoring Type | BLEU | Entity-F1 | Re@7 |
|---|---|---|---|
| Few-shot($rank_{sr-bleu}$) | 7.14 | 31.46 | 88.03 |
| Few-shot($rank_{sr}$) | 6.03 | 29.40 | 84.07 |

Table 10: The accuracy of the self-reported ChatGPT scores

### F.3 Prompt for ChatGPT

Since the ChatGPT API does not directly provide the generation probabilities of responses, which are required in our method for constructing feedback and entity scoring, we make slight adaptations. For negative sample selection, we construct prompts to elicit ChatGPT to generate responses along with their corresponding confidence scores as approximations of generation probabilities. We further select hypothesis responses with low confidence scores but high actual quality as negative examples. Regarding entity scoring, we employ a similar approach by constructing prompts to assess the relevance of each (entity, response) pair. The specific prompts used in this process are shown in Table 11 and Table 12. It should be noted that the prompts used in the zero-shot and few-shot scenarios only differ in the presence of examples, and therefore the few-shot prompts are not separately displayed.

You are a task-oriented dialogue chatbot. Your initial priority is to understand the user's intent within the current user input, taking into account the dialogue history. Subsequently, you need to select the relevant information from the knowledge base that aligns with this intent. Finally, generate concise response by incorporating the current user input and the selected information from the knowledge base. Additionally, you need provide a confidence score for each response to indicate the level of certainty associated with it. The confidence score falls within the range of 0.0 to 1.0, denoted as a decimal. The output format of responses follows the structure:

Response: [Generated response]

Confidence: [Confidence score]

**Knowledge base**

1. name charlie chan, address regent street city centre, area centre, domain restaurant, food chinese, phone 01223361763, postcode cb21db, pricerange cheap, type restaurant.

2. name alexander bed and breakfast, address 56 saint barnabas road, area centre, domain hotel, internet yes, parking yes, phone 01223525725, postcode cb12de, pricerange cheap, stars 4 star,

3. name restaurant one seven, address de vere university arms regent street city centre, area centre, food british, phone 01223337766, postcode cb21ab, pricerange moderate, type restaurant.

4. \*\*\*

**Dialogue history**

[user]: are there any restaurants that serve proper british food in town ?

[sys]: oh yes quite a few . which part of town will you be dining in ?

**User input**

[user]: west , if possible .

**Response:**

Table 11: Response prompt for ChatGPT.

You are required to assign relevance scores to each entity-response pair in the input. These scores should range from 0.0 to 1.0 and maintain the order based on the input sequence and the total number of entities in the knowledge base. The output format should follow the pattern:

Score: [relevance-score1, relevance-score2, ...]

**Knowledge base**

1. name charlie chan, address regent street city centre, area centre, domain restaurant, food chinese, phone 01223361763, postcode cb21db, pricerange cheap, type restaurant.

2. name alexander bed and breakfast, address 56 saint barnabas road, area centre, domain hotel, internet yes, parking yes, phone 01223525725, postcode cb12de, pricerange cheap, stars 4 star,

3. name restaurant one seven, address de vere university arms regent street city centre, area centre, food british, phone 01223337766, postcode cb21ab, pricerange moderate, type restaurant.

4. \*\*\*

**Response:**

we have three: graffiti , saint john ' s chop house , and traveller ' s rest .

**Score:**

Table 12: Entity scoring prompt for ChatGPT.