# OpenReview forum: "Dual-Feedback Knowledge Retrieval for Task-Oriented Dialogue Systems"
_EMNLP/2023/Conference — EMNLP 2023 Main_

### Official Review · Reviewer_gvAr · 2023-08-04

**Soundness:** 4

**Excitement:**

3: Ambivalent: It has merits (e.g., it reports state-of-the-art results, the idea is nice), but there are key weaknesses (e.g., it describes incremental work), and it can significantly benefit from another round of revision. However, I won't object to accepting it if my co-reviewers champion it.

**Paper Topic And Main Contributions:**

This paper proposed a Task-Oriented Dialogue System with retriever-generator architecture. The retriever to retrieve pertinent knowledge and a generator to generate system responses. Due to the lack of retriever training labels, for the optimization of retriever, this paper introduced a dual-feedback mechanism where the generator produced positive and negative feedback as pseudo labels. The experiments on three public benchmarks demonstrated the effectiveness of the proposed method.

**Questions For The Authors:**

1.	During the training process, are the retriever and generator jointly optimized with the final loss function?

**Reasons To Accept:**

1.	This paper utilized generator to produce positive labels as pseudo labels of retrieved entities for optimization, which addressed the lack of retriever training labels.
2.	This paper further proposed negative feedback prevent the retriever from being misled by inaccurate information.
3.	This paper conducted extensive experiments and analysis which demonstrated the effectiveness of proposed methods


**Reasons To Reject:**

1.	The paper mentioned that the introduction of negative feedback is to solve the problem of positive feedback assimilating incorrect knowledge from the entities, but the negative loss function in Section 3.2 might not meet this purpose well. It would be better to explain it more fully.
2.	It was insufficient that evaluating natural language generation (NLG) systems with only word-overlap based metrics such as BLEU. It would be better to use semantic-based metrics such as BERTSCORE.
3.	In Section 5.4, paper prompted ChatGPT to obtain entity scores. it was different from the proposed method which utilizing conditional log probability as entity scores. It was not enough to verify the compatibility of proposed positive feedback on LLMs.


**Reproducibility:**

4: Could mostly reproduce the results, but there may be some variation because of sample variance or minor variations in their interpretation of the protocol or method.

**Reviewer Confidence:**

4: Quite sure. I tried to check the important points carefully. It's unlikely, though conceivable, that I missed something that should affect my ratings.

---

> ### Author Rebuttal · Authors · 2023-08-28
>
> Thank you for your thorough reviews and insightful suggestions.
>
> **Q1: Explanation of the effectiveness of the negative loss function.**
>
> **A1:** We appreciate your observation regarding the oversight in formulating the negative loss function. In particular, we inadvertently omitted the negative sign after subtracting the two $D_{KL}(\cdot)$ terms. The correct negative loss function should be:
> \begin{equation}
> L_{\text{neg}} = \max\left(0, -(D_{\text{KL}}(G_{t}^{-}, S_{t}) - D_{\text{KL}}(G_{t}, S_{t})) + \eta\right).
> \end{equation}
>
> In the actual implementation (Source code: `Dual-Feedback-TOD/model/FiD_ToD_FullKB.py`: line 645), this function is instantiated using `torch.nn.MarginRankingLoss()` *(y=1)*. Thus, this minor error has no bearing on the experimental results. We will rectify this oversight when revising the paper.
>
> **Q2: Choice of metrics for evaluating natural language generation (NLG).**
>
> **A2:** We concur that BLEU is not an ideal metric for assessing tasks involving NLG, and semantic-oriented metrics like BERTSCORE could be considered as an alternative measure. However, we opted for BLEU due to its computational efficiency, whereas semantic-focused metrics such as BERTSCORE might require additional model training. This introduces a notable challenge in acquiring task-specific pairwise training data for dialogues. In our future research, we will delve into exploring semantic-based metrics such as BERTSCORE.
>
> **Q3: The compatibility of our proposed dual-feedback mechanism with LLMs.**
>
> **A3:** This paper aims to introduce a retriever-generator architecture that combines a retriever for retrieving relevant knowledge and a generator for producing system responses. Given the absence of retriever training labels, our primary focus is on leveraging feedback from the generator as pseudo-labels to train the retriever. The experimentation with LLMs aims to showcase the compatibility of our proposed dual-feedback mechanism with these models. We opted for the closed-source ChatGPT over other open-source language models due to its relatively lower demand for computing resources. However, in the revised version of the paper, we will expand our experimentation to include open-source LLMs as well.
>
> **Q4: Detailed description of the training process.**
>
> **A4:** We provide a more comprehensive description of the training process below. During the initial training phases, we incorporate a warm-up period for the generator, facilitating dependable feedback for the retriever. Subsequent to this phase, both the retriever and generator are subjected to joint training until reaching convergence. It's important to emphasize that we carry out negative sampling at each training step, ensuring consistent information updates from the generator to the retriever. We intend to include this detailed training process description in the revised version of the paper.
>
> Thanks again for your helpful feedback and suggestions. Please let us know if you have any further questions, and we are happy to discuss further.

---

### Official Review · Reviewer_iQGR · 2023-08-05

**Soundness:** 4

**Excitement:**

3: Ambivalent: It has merits (e.g., it reports state-of-the-art results, the idea is nice), but there are key weaknesses (e.g., it describes incremental work), and it can significantly benefit from another round of revision. However, I won't object to accepting it if my co-reviewers champion it.

**Paper Topic And Main Contributions:**

This paper proposes a new retriever-generator training architecture for task-oriented dialog systems. This architecture is able to train the retriever using positive and negative feedback signals from the generator. The authors conducted experiments on three publicly available datasets and obtained good performance.

**Questions For The Authors:**

1. How long does the training process take?

**Reasons To Accept:**

1. This method achieves good performance on three publicly available datasets. In addition, the authors conducted a variety of complementary experiments to demonstrate the validity of their method.
2. Although the retriever-generator architectures have become a popular trend in the field of dialog systems, most of the current work still combines these two modules in a pipelined form. This work, on the other hand, designs a method to pass positive and negative feedback from the generator to the retriever, which is innovative and can be generalized to other approaches that employ retriever-generator architectures.
3. The source code is provided.

**Reasons To Reject:**

1. Generating negative samples during training may affect efficiency.
2. There is a lack of detailed description of the training process, e.g., when end-to-end training starts and how often negative samples are sampled.

**Reproducibility:**

4: Could mostly reproduce the results, but there may be some variation because of sample variance or minor variations in their interpretation of the protocol or method.

**Reviewer Confidence:**

4: Quite sure. I tried to check the important points carefully. It's unlikely, though conceivable, that I missed something that should affect my ratings.

---

> ### Author Rebuttal · Authors · 2023-08-28
>
> Thank you for your valuable suggestions and insightful questions.
>
> **Q1: Detailed description of the training process.**
>
> **A1:** We provide a more comprehensive description of the training process below. During the initial training phases, we incorporate a warm-up period for the generator, facilitating dependable feedback for the retriever. Subsequent to this phase, both the retriever and generator are subjected to joint training until reaching convergence. It's important to emphasize that we carry out negative sampling at each training step, ensuring consistent information updates from the generator to the retriever. We intend to include this detailed training process description in the revised version of the paper.
>
> **Q2: Regarding the training efficiency.**
>
> **A2:** We acknowledge that the inclusion of sampling operations during training does indeed extend the overall training duration. Nonetheless, it's important to note that these sampling operations are exclusively carried out within the training phase and therefore do not impact the inference time. Moreover, through our evaluation on the validation set during training, we observe that our dual-feedback approach consistently outperforms the baseline in terms of results achieved within the same number of training steps. This indicates that our approach can attain favorable results earlier in the training process.
>
> Thanks again for your review! Please let us know if you have any further questions.

---

### Official Review · Reviewer_ctRL · 2023-08-05

**Soundness:** 4

**Excitement:**

4: Strong: This paper deepens the understanding of some phenomenon or lowers the barriers to an existing research direction.

**Paper Topic And Main Contributions:**

The paper explores how to train the retriever in a retrieve-and-generate approach for knowledge-grounded dialogue systems, assuming no access to explicitly annotated data. Specifically, the proposed approach first assigns to the retrieved entities a score calculated as the normalized conditional (on context and entities) log probability of the generated sequence according to the model (i.e. a pretrained LM). The retriever is then fitted to these scores via KL divergence. Furthermore, the scores of entities that result in generator samples with low BLEU score but high generator probability, is treated as negative feedback for the retriever.

The paper presents experimental results on MultiWoz, Stanford Multi-Domain Dataset, and CamRest, on two settings employing a session-level and dataset-level knowledge base correspondingly. The approach is shown to outperform recent work on BLEU and Entity-F1, and is accompanied by an ablation study,  a comparison to other training methods for the retriever, and an analysis on how to best select the negative samples.


**Questions For The Authors:**

A. Please provide more details regarding the  zero-shot and few-shot experiments, e.g. how many ChatGPT samples were used to train the retriever? Was any further study done to support the accuracy of the self-reported ChatGPT scores? Could useful scores be obtained by estimating the probability of the ChatGPT outputs through a different pretrained language model?


**Reasons To Accept:**

- The approach shows good performance, outperforming previous work on three dataset across two settings.
- The provided analysis and comparisons further support the proposed method.
- The paper is for the most part easy to follow and clear to understand.


**Reasons To Reject:**

- The zero-shot and few-shot experiments are somewhat underexplained.


**Reproducibility:**

4: Could mostly reproduce the results, but there may be some variation because of sample variance or minor variations in their interpretation of the protocol or method.

**Reviewer Confidence:**

3: Pretty sure, but there's a chance I missed something. Although I have a good feel for this area in general, I did not carefully check the paper's details, e.g., the math, experimental design, or novelty.

**Typos Grammar Style And Presentation Improvements:**

- In Figure 1, perhaps include the price value as its unclear alone why Chesterton should be preferred.
- In Tables 1 and 2, please  indicate the size of the underlying models (in number of parameters) in each approach, to help the reader make a fair comparison between the methods. Specifically, in Table 2, one would assume that Q-TOD is using the base model, but this should be indicated.

---

> ### Author Rebuttal · Authors · 2023-08-28
>
> Thank you for your thoughtful reviews and valuable suggestions!
>
> **Q1: More details regarding the zero-shot and few-shot experiments.**
>
> **A1:** Regarding the zero-shot and few-shot experiments, our retriever underwent training on the complete MultiWOZ dataset, with a total of 2877 dialogues. Furthermore, in the few-shot context, we create prompts to simulate three main knowledge retrieval scenarios: knowledge base (KB) containing entities that align with user intent, KB containing similar entities for recommendations, and KB lacking similar entities and leading to query failures. We intend to incorporate these particulars during the revision of our paper.
>
> **Q2: The accuracy of the self-reported ChatGPT scores.**
>
> **A2:** This paper aims to introduce a retriever-generator architecture that combines a retriever for retrieving relevant knowledge and a generator for producing system responses. Given the absence of retriever training labels, our primary focus is on leveraging feedback from the generator as pseudo-labels to train the retriever. The experimentation with LLMs aims to showcase the compatibility of our proposed dual-feedback mechanism with these models.
>
> As indicated in the table below, relying solely on self-reported (sr) scores for constructing feedback leads to performance degradation compared to the approach proposed in the paper. This, to some extent, validates the presence of inaccuracies in self-reported scores. However, quantifying the precision is challenging due to the absence of ground truth labels for ChatGPT's outputs. Furthermore, our approach combines self-reported scores with BLEU-based ranking for selecting negative samples, thereby mitigating the unfavorable effects that may arise from these inaccuracies.
>
> | Few-Shot                | BLEU | Entity-F1 | Re@7   |
> |-----------------------|------|-----------|--------|
> |$rank_{sr-bleu}$| 7.14 | 31.46     | 88.03  |
> | $rank_{sr}$      | 6.03 | 29.40     | 84.07  |
>
>
> **Q3: Suggested modifications.**
>
> **A3:** Thank you for the feedback. We will certainly take your suggestions into consideration during the paper revision process.
>
> Thanks again for your review! Please let us know if you have any further questions, and we are happy to discuss further.

---

### Meta-Review · Area_Chair_2zpj · 2023-09-25

**Recommendation:** 5

**Metareview:**

An interesting coupling of knowledge retriever and dialog generator to improve the quality of the retrieved knowledge to be incorporated in the generated dialog response. The paper is sound and shows clear evidence of the efficiency of the proposed method.

---

### Decision · Program_Chairs · 2023-10-07

**Decision:**

Accept-Main

**Comment:**

An interesting coupling of knowledge retriever and dialog generator to improve the quality of the retrieved knowledge to be incorporated in the generated dialog response. The paper is sound and shows clear evidence of the efficiency of the proposed method.